# Quantifying accuracy and heterogeneity in single-molecule super-resolution microscopy

Hesam Mazidi[1], Tianben Ding [1], Arye Nehorai[1] & Matthew D. Lew [1]✉

The resolution and accuracy of single-molecule localization microscopes (SMLMs) are routinely benchmarked using simulated data, calibration rulers, or comparisons to secondary imaging modalities. However, these methods cannot quantify the nanoscale accuracy of an arbitrary SMLM dataset. Here, we show that by computing localization stability under a well-chosen perturbation with accurate knowledge of the imaging system, we can robustly measure the confidence of individual localizations without ground-truth knowledge of the sample. We demonstrate that our method, termed Wasserstein-induced flux (WIF), measures the accuracy of various reconstruction algorithms directly on experimental 2D and 3D data of microtubules and amyloid fibrils. We further show that WIF confidences can be used to evaluate the mismatch between computational models and imaging data, enhance the accuracy and resolution of reconstructed structures, and discover hidden molecular heterogeneities. As a computational methodology, WIF is broadly applicable to any SMLM dataset, imaging system, and localization algorithm.

[1] Department of Electrical and Systems Engineering, Washington University in St. Louis, St. Louis, MO 63130, USA. ✉email: mdlew@wustl.edu

Single-molecule localization microscopy (SMLM) has become an important tool for resolving nanoscale structures and answering fundamental questions in biology[1–3] and materials science[4,5]. SMLM uses repeated localizations of blinking fluorescent molecules to reconstruct high-resolution images of a target structure. In this way, quasi-static features of the sample are estimated from noisy individual images captured from a fluorescence microscope. These quantities, such as fluorophore positions (i.e., a map of fluorophore density), blinking on times, emission wavelengths, and orientations, influence the random blinking events that are captured within an SMLM dataset. By using a mathematical model of the microscope, SMLM reconstruction algorithms seek to estimate the most likely set of fluorophore positions and brightnesses (i.e., a super-resolution image) that is consistent with the observed noisy images.

A key question left unresolved by existing SMLM methodologies is: How well do the SMLM data, i.e., the images of blinking single molecules (SMs), support the super-resolved image produced by an algorithm? That is, what is our statistical confidence in each localization? Intuitively, one's interpretation of an SMLM reconstruction could dramatically change by knowing how trustworthy each localization is.

Existing metrics for assessing SMLM image quality can be categorized broadly into two classes: those that require knowledge of the ground-truth positions of fluorophores (e.g., Jaccard index and imaging DNA calibration rulers)[6–9], and those that operate directly on SMLM reconstructions alone, possibly incorporating information from other measurements (e.g., diffraction-limited imaging)[10–12].

While these methods are able to provide summary or aggregate measures of performance, none of them directly measure the accuracy of individual localizations in an arbitrary SMLM dataset. Such knowledge is critical for harnessing fully the power of SMLM for scientific discovery.

Here, we leverage two fundamental insights of the SMLM measurement process: (1) we possess highly accurate mathematical models of the imaging system, and (2) we know the precise statistics of the noise within each image. Our proposed computational method, termed Wasserstein-induced flux (WIF), uses this knowledge to assess quantitatively the confidence of each individual localization within an SMLM dataset without knowledge of the ground-truth. Localizations with high confidences indicate that their positions and brightnesses are accurate and precise and thus lead to improved image resolution, while those with low confidences exhibit inaccuracies, poor precision, or both and lead to poor resolution and image artifacts. With these confidences in hand, the experimenter may filter unreliable localizations from SMLM images without removing accurate ones necessary to resolve fine features. These confidences may also be used to detect mismatches in the mathematical imaging model that create image artifacts[13], such as misfocusing of the microscope, dipole-induced localization errors[14], and the presence of optical aberrations[15,16].

## Results

**Measuring localization confidence via Wasserstein-induced flux.** In contrast to Poisson shot noise, which degrades the achievable measurement precision, modeling errors arising from the sample, microscope, and SMLM software (e.g., isotropic vs. dipole-like emission, dense SM blinking, optical aberrations, and sub-optimal software parameters) can cause inaccuracies per localization beyond intrinsic errors from shot noise, thereby degrading image resolution and introducing imaging artifacts. As these confounding effects are often hidden or difficult to detect, we must somehow estimate the degree of uncertainty or confidence of each localization. Our key observation is that unreliable localizations, i.e., a set of SM position and brightness estimates, are unstable upon a well-chosen computational perturbation[17]. To leverage this mathematical principle, we develop a computational imaging algorithm consisting of two stages. In the first stage, we perturb each localization (Fig. 1a) by dividing its photons among eight adjacent positions with equal brightnesses (Fig. 1b, "Methods"). Next, we solve a regularized transport problem, which basically computes how the perturbed sources move while minimizing the regularized negative log likelihood (Fig. 1b, "Methods"). In order to estimate the stability of a localization, we measure the degree of photon flux that returns toward the original localization from the perturbed positions (Fig. 1, "Methods"). Normalized from −1 (least confidence) to 1 (highest confidence), we call this quantity WIF, as it has an elegant connection to Wasserstein gradient flows (Supplementary Note 1)[18]. Consider when half of the perturbed sources exhibit transport trajectories that equally converge and diverge from the original estimate, and thus contribute a net zero to the flux, while the remaining converge toward the original localization: in this case, WIF approaches 0.5, and our confidence in the localization is half as certain. Thus, we interpret WIF = 0.5 as a threshold for detecting inaccurate localizations (Supplementary Fig. 2), but others may be chosen depending on a specific imaging task.

Quantifying subtle model mismatches in SMLM is a challenging problem. For 2D SMLM, the fitted width $\hat{\sigma}$ of the standard point-spread function (PSF) is commonly used; if $\hat{\sigma}$ is significantly smaller or larger than the expected width of the fitted PSF, then the corresponding localization is deemed to have low confidence. To test this metric, we analyzed images of an SM and two closely spaced molecules (70 nm separation) whose images overlap. In both scenarios, SMLM algorithms always detect only one molecule, such that in the latter, the estimated positions exhibit significant deviations from the true ones (Fig. 1c, d). However, the distributions of $\hat{\sigma}$ in both cases are virtually identical, suggesting that simple perturbations to the PSF, e.g., a change in $\hat{\sigma}$, are insufficient for detecting errors due to overlapping molecules. More fundamentally, mismatches in SMLM between model and measurement generally cannot be quantified via simple image-based features such as PSF width. For example, when localizing a dipole emitter (e.g., a fluorescent SM) defocused by 200 nm, its anisotropic emission pattern induces a significant bias in the estimated positions. The distribution of fitted widths is noisy due to photon-shot noise and broadening of the PSF (Fig. 1e). Interestingly, these fitted widths are comparable to those of a dim molecule with an isotropic emission pattern, whose localizations have no systematic bias (Fig. 1f). In contrast, we see that when the estimated localizations are close to the ground-truth positions, their estimated confidences or WIFs are concentrated close to 1 (Fig. 1c, f). On the other hand, for inaccurate estimates, localization confidences become significantly smaller, indicating their unreliability (Fig. 1d, e). Note that knowledge of the ground-truth molecule location is not needed to compute these confidence values.

WIF computes the consistency of a set of localizations with respect to the raw SMLM data and a given PSF model, which can be estimated directly from experimental images of SMs or from a calibration dataset (see "Methods"). To assess the performance of WIF for detecting model mismatch, and thus localization errors, we simulate images of fluorescent molecules, generated using a vectorial image-formation model[19], with perturbations to various hidden sample parameters such as defocus, molecular rotational diffusion, and sample refractive index (Supplementary Note 6 and "Methods"). To compute WIF, we set the 2D PSF model to that of an isotropic emitter at zero defocus. We observe that WIF provides a consistent and reliable measure of localization

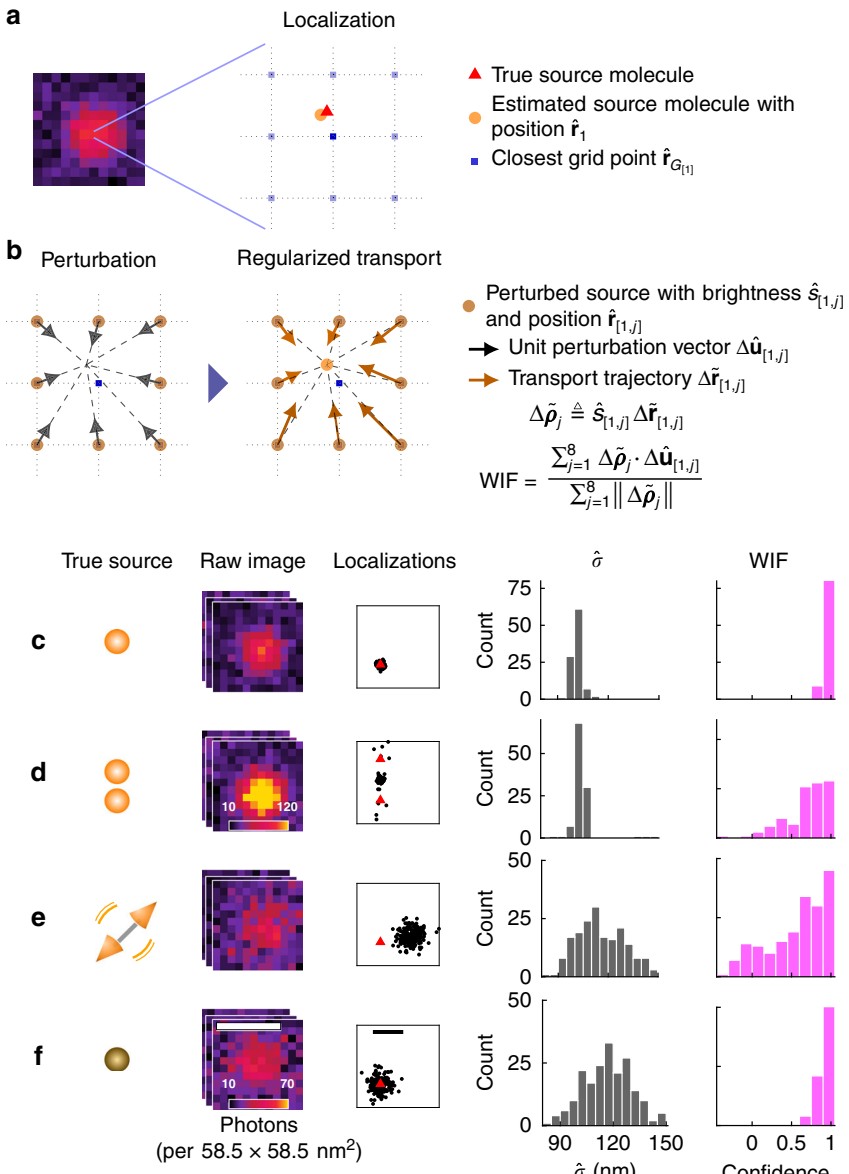

**Fig. 1 Quantifying confidence in single-molecule localization microscopy (SMLM). a** Left: Simulated image of a single molecule (SM, position denoted by red triangle) with isotropic emission. Right: Localization (orange circle) refers to a position $\hat{\mathbf{r}}_1$ and brightness $\hat{s}_1$ estimate returned by an SMLM algorithm. **b** Proposed confidence quantification framework. Localizations are represented as positions and brightnesses referenced to a grid without loss of generality (pale blue squares, "Methods"). Left: A perturbation divides the photons of each estimated source molecule equally across eight neighboring grid points with brightnesses $\hat{s}_{[1,j]}$ at positions $\hat{\mathbf{r}}_{[1,j]}$. Middle: The perturbed source molecules are fed to a regularized transport optimization algorithm that minimizes a regularized negative log likelihood using its own PSF model, resulting in transport trajectories $\Delta\tilde{\mathbf{r}}_{[1,j]}$. Wasserstein-induced flux (WIF) measures the normalized amount of inward photon flux from the neighboring perturbed source molecules, taking values from −1 (least confidence) to 1 (highest confidence). **c–f** Examples of localizing and quantifying confidence. **c** 100 simulated images of an isotropic, in-focus SM analyzed by ThunderSTORM (TS). Scatter plot: localizations (black dots) and true positions of the molecules (red triangles). Gray histogram: fitted widths of the PSF ($\hat{\sigma}$) estimated by TS. Magenta histogram: estimated WIF confidences using the proposed method. **d** Similar to (**c**) but for two molecules separated by 70 nm. **e** Similar to (**c**) but for a dipole-like molecule defocused by 200 nm. **f** Similar to (**c**) but for a dim isotropic molecule in focus. Color bars: **a**, **c**, **d** and **e**, **f** photons per 58.5 × 58.5 nm². Scale bars: **a** and **f** left: 500 nm, **f** right: 50 nm.

confidence in the presence of experimental mismatches for a broad range of molecular brightness (Supplementary Figs. 4–8 and Supplementary Note 6). Notably, confounding factors (e.g., a defocused dipole-like emitter) may cause estimates of PSF width to appear unbiased, while our WIF metric consistently detects these image distortions, yielding small confidence values (Supplementary Figs. 7, 8) and producing a quantitative, interpretable measure of image trustworthiness. Moreover, in the case of 3D SMLM, we have found that WIF reliably detects

errors in axial localization caused by index mismatch-induced PSF aberrations (Supplementary Fig. 9).

Next, we characterize the impact of signal-to-noise ratio (SNR) on WIF's sensitivity to detect and quantify position errors. Here, we define (peak) SNR as the ratio of the number of photons ($s_{sig}$) in the brightest pixel of a PSF to the square root of the sum of $s_{sig}$ and the detected background photons in that pixel[20]. Intuitively, we expect that as SNR decreases, the likelihood landscape becomes increasingly rough and uninformative; severe noise will

cause the regularized transport process to prefer sparser solutions whose transport trajectories return toward the position of the original estimate. In particular, position errors that are comparable to the achievable localization precision, especially at low SNRs, may not be detected by WIF (Supplementary Fig. 10a). However, when the position error is beyond three times the achievable localization precision (error >3× the square root of the Cramér−Rao bound), WIF is able to detect inaccurate localizations with accuracy greater than 80% (Supplementary Fig. 10b, c). We also measured how well WIF detects errors in expected brightness due to aberrations in the PSF; WIF confidences fall significantly below 0.5 as the brightness error becomes much larger than the achievable brightness precision (Supplementary Fig. 11). We also observe that WIF is more sensitive to brightness errors due to astigmatism versus defocus (Supplementary Fig. 11a, c). This effect may arise from the higher effective SNR of the astigmatic images and the asymmetric distortion of the astigmatic PSF (Supplementary Fig. 11g, h), both of which contribute to transport trajectories that significantly deviate from the original localization (Supplementary Fig. 11e, f). We stress that WIF detects these errors without prior assumptions on the source of the error or statistical averaging over many localizations.

Since WIF depends on the log likelihood function, the statistical distribution of our confidence estimates (e.g., those shown in Fig. 1) depends on SNR and the degree of model mismatch or error in the original localization. That is, both shot noise and localization errors will affect individual WIF estimates, as well as their mean and standard deviation over many measurements (Fig. 1c−f). Interestingly, we see that when the SNR is low, the width of the WIF distribution is wider for inaccurate localizations than that for accurate ones (Fig. 1e vs. Fig. 1f). Therefore, we can use estimates of WIF's stability or precision, especially at low SNRs, to further improve WIF's error-detection capability. Intuitively, if a localization is unreliable under a low SNR, its likelihood landscape should be locally rough and thus exhibit various local minima. Thus, if we allow perturbed sources to explore various regions around the estimated source (by testing a variety of optimization constraints), then we expect heterogeneous WIF estimates for inaccurate localizations under low SNR. In contrast, we expect WIF to be less sensitive to changes in the constraint for both accurate and inaccurate localizations under high SNR (Supplementary Figs. 12, 13).

We estimate WIFs for a range of constraints (or equivalently regularizer strengths $v$) and compute the median and median absolute deviation (MAD) statistics (Supplementary Note 3), which are more robust to outliers than mean and standard deviation. A confidence interval then can be constructed for the estimated WIF by approximating the WIF variance as 1.48× MAD. For the largely inaccurate localizations of a fixed, defocused dipole (Fig. 1e), remarkably, we find that using thresholds on both the median and (MAD) standard deviation of WIF (median threshold of 0.5 and std. dev. threshold of 0.1) can detect inaccurate localizations with an accuracy of 84% (Supplementary Fig. 14). We note that although using the standard deviation of WIF improves detecting inaccurate localizations at low SNRs, it comes with a cost of computing multiple WIFs, which could be computationally expensive. In addition, at typical SMLM SNRs, WIFs computed based on a single constraint adequately quantify localization inaccuracies.

Next, we consider the behavior of WIF for two closely located molecules at various separation distances. When their separation distance is small (70 nm), the localizations' WIF values are significantly smaller than one (Supplementary Fig. 15a). These low values directly arise from diverging transport trajectories of the perturbed sources (Supplementary Fig. 15h), signaling that

the original localizations have large biases. On the other hand, when the molecules are well separated (280 nm), the trajectories return toward the original localizations, and thus we observe high WIF confidences (Supplementary Fig. 15i, j).

To consider more complex scenarios, we analyze a typical SMLM dataset of stochastically blinking molecules simulated using an ideal imaging model, i.e, with no mismatch (Supplementary Note 8 and "Methods"). We propose average−confidence $WIF_{avg}$ as a novel metric for quantifying the collective accuracy of localizations returned by an algorithm: $WIF_{avg} \triangleq \frac{1}{N} \sum_{i=1}^{N} c_i$, where $N$ and $c_i$ denote the number of localizations and the confidence of the $i$th localization, respectively. As a demonstration at various blinking densities (number of molecules per μm²; see "Methods"), we compare the performance of three algorithms, RoSE[7], a sparsity-promoting maximum likelihood estimator (MLE); FALCON[21], another sparse deconvolution algorithm with a different formulation; and ThunderSTORM (TS)[22], which uses local peak detection followed by MLE (Supplementary Fig. 16a, b). For RoSE, FALCON, and TS, we observe excellent agreement between $WIF_{avg}$ and the Jaccard index ("Methods"), which identifies accurate localizations using the ground-truth molecule positions, for densities as high as 5 mol μm⁻² (Supplementary Fig. 16c). In addition, by removing localizations with poor confidence, we gain a significant increase in detection precision as high as 180% for TS and 23% for RoSE (density = 9 mol μm⁻², Supplementary Fig. 16d). Remarkably, these improvements come with a negligible loss in detection performance (13% drop in recall in the worst case) across all densities for all algorithms (Supplementary Fig. 16e). These observations consistently hold for 3D datasets as well (Supplementary Fig. 17). We further used WIF to construct a confidence map of localizations for a synthetic benchmark high-density (HD) SMLM dataset[6]. In contrast to other error metrics, the WIF confidence map enables us to discriminate specific SM localizations that are trustworthy, while also assigning low confidence values to those that are not, thereby maximizing the utility of SMLM datasets without throwing away useful localizations (Supplementary Fig. 18).

**Calibrating and validating WIF using SMLM of microtubules.** A super-resolution dataset often contains well-isolated images of molecules, e.g., after a significant portion of them are bleached. These images can therefore serve as a useful internal control, taken under realistic conditions, to assess the performance of a PSF model as well as SMLM algorithms themselves on a particular dataset. As a practical example, we examine an SMLM dataset of blinking Alexa Fluor 647-labeled microtubules ("Methods"). We randomly selected 600 images of bright molecules sampled over the entire field of view (Fig. 2a). We used an ideal PSF model to localize these molecules using RoSE, but found that the mean confidence of these localizations is notably small ($WIF_{avg} = -0.36$), implying the presence of significant aberrations and PSF model mismatch (Supplementary Fig. 19). We therefore calibrated our physics-based PSF model in both the localization and confidence measurement steps and re-analyzed the data ("Methods"). After calibration, the estimated confidences of RoSE's localizations show a notable average increase of 0.79 ($WIF_{avg} = 0.43$). We also observe a rather broad distribution of confidences, suggesting that optical aberrations, such as defocus, vary throughout the structure (Supplementary Fig. 19). We further observe that RoSE's use of this calibrated PSF produces localizations with higher confidence values ($WIF_{avg} = 0.43$) compared to TS's use of an elliptical Gaussian PSF ($WIF_{avg} = 0.15$) (Fig. 2a). The higher average confidence score for RoSE suggests that it should recover the underlying structure with greater accuracy compared to TS.

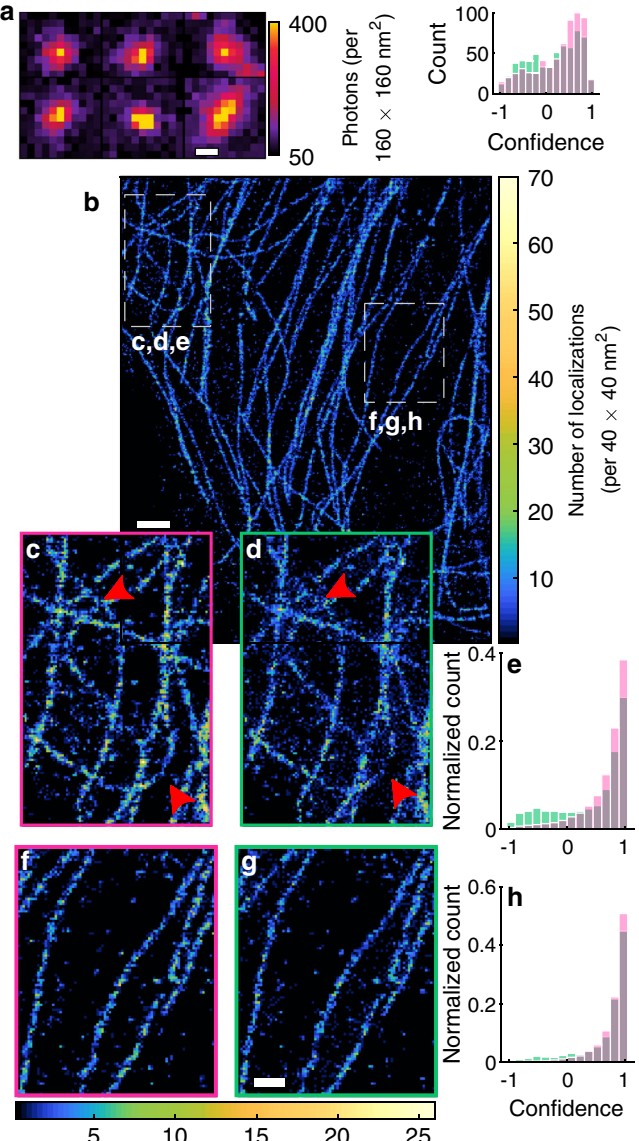

**Fig. 2 Comparison of 2D SMLM algorithms on experimental images of Alexa Fluor 647-labeled microtubules. a** Left: isolated images of Alexa Fluor 647 molecules. Right: localization confidences for 600 isolated molecules using RoSE (red) and TS (green). **b** SMLM image of microtubules recovered by RoSE. **c, d** Enlarged top-left region in (**b**) for RoSE and TS, respectively. **e** Histogram of confidences corresponding to localizations in (**c**) and (**d**) for RoSE (red) and TS (green), respectively. **f, g** Similar to (**c, d**) but for the middle-right region in (**b**). **h** Similar to (**e**) but for localizations in (**f**) and (**g**). Color bars: **a** photons detected per $160 \times 160$ nm$^2$, **b** number of localizations per $40 \times 40$ nm$^2$. Scale bars: **a** 500 nm, **b** 1 µm, and **g** 500 nm.

We confirm the reliability of our WIF confidence metric, in the absence of the ground-truth, through its correlation with the perceived quality of the super-resolution reconstructions (Fig. 2b). We expect more confident localizations to result in an image with greater resolution, whereas localizations with poor confidence should fail to resolve fine details and could potentially distort the structure. Within a region containing a few parallel and well-separated microtubules, we see similar confidences for both algorithms (Fig. 2h) resulting in images of similar quality (Fig. 2f, g). Conversely, for a region with intersecting microtubules, we observe marked qualitative and quantitative differences between the two reconstructions (Fig. 2c, d). RoSE is able to resolve structural details near the intersections, while the TS image

contains missing and blurred localizations near the crossing points. Moreover, RoSE recovers the curved microtubule faithfully, whereas TS fails to reconstruct its central part (lower red arrow in Fig. 2c, d). Quantitatively, RoSE exhibits significantly greater confidence in its localizations compared to TS, which shows negative confidences for an appreciable number of localizations (Fig. 2e). This confidence gap is likely caused by hidden or unmodeled parameters within the data, such as high blinking density.

Three-dimensional SMLM datasets, especially those obtained using engineered PSFs, pose several challenges for localization algorithms, including mismatches between ideal and experimental PSFs, frequently overlapping SM images arising from dense 3D structures, and spatially and temporally varying background (Fig. 3a). Here, we further validate the usefulness of WIF in quantifying the accuracy of 3D PSF models and algorithms. We first built a 3D PSF model from a $z$-stack of bright fluorescent beads imaged with the Double-Helix PSF (DHPSF)[23], using optimal-transport (OT)-based interpolation to align multiple beads in a field-of-view ("Methods"). We found that OT interpolation substantially improves WIFs compared to pupil-based phase retrieval (Supplementary Note 9) and that WIF is correlated with the accuracy of the beads' estimated positions (Supplementary Fig. 21). Our experimentally derived PSF also accurately modeled isolated images of Alexa Fluor 647 molecules attached to microtubules (mean confidence = 0.71, Supplementary Fig. 22b, c).

Next, we analyzed 3D SMLM images of a complex microtubule network spanning a 1-µm axial range (Fig. 3a) using two algorithms, RoSE and Easy-DHPSF[24]. Interestingly, WIF revealed a degradation in RoSE's performance when DHPSF images overlapped frequently (mean and median WIFs of 0.42 and 0.52 respectively, Supplementary Fig. 22f). We inferred that model mismatch induced fitting instability within RoSE, and we optimized its iterative fitting scheme to significantly increase WIF performance (mean and median WIFs of 0.69 and 0.8 respectively, Supplementary Fig. 22g, Supplementary Note 4).

We compared the performance of RoSE and Easy-DHPSF by randomly selecting 500 raw SMLM frames (corresponding to 2,425 of the 48,445 localizations plotted in Fig. 3c, d) and computing the corresponding WIFs. Notably, we see that RoSE has appreciably higher WIFs (mean and median of 0.66 and 0.78) compared to Easy-DHPSF (mean and median of 0.45 and 0.61) (Fig. 3b). Indeed, these higher WIFs are consistent with the superior perceived quality of RoSE's reconstruction (Fig. 3d) compared to that of Easy-DHPSF (Fig. 3c). In particular, in a region with an isolated microtubule (region 1), both algorithms reveal the circular cross-section of the microtubule (Fig. 3e); however, RoSE's localizations exhibit slightly higher precision along the $z$ axis (Supplementary Fig. 23). Moreover, in a dense region (region 2), multiple crossing microtubules are clearly resolved in RoSE's localizations (Fig. 3f). In terms of WIF, we see that RoSE's WIFs in region 1 (mean 0.71 and median 0.87) slightly outperform those of Easy-DHPSF (mean 0.7 and median 0.76), while in region 2, RoSE's WIF distribution (mean 0.62 and median 0.74) has dramatically better confidence than that of Easy-DHPSF (mean 0.38 and median 0.43). Further, we observe that regions with high WIF scores consistently show better image quality (Supplementary Fig. 24); for example, the reconstructed microtubule in region 1 appears to be narrower than those in region 2, which is reflected in the superior WIFs for both algorithms (Fig. 3).

Overall, these data indicate that WIF detects sub-optimal PSF models and algorithms directly from experimental SMLM data, obviating the need to know the ground-truth structure. Unlike correlation-based resolution metrics, relatively few imaging frames are required (e.g., only a few hundred) to meaningfully

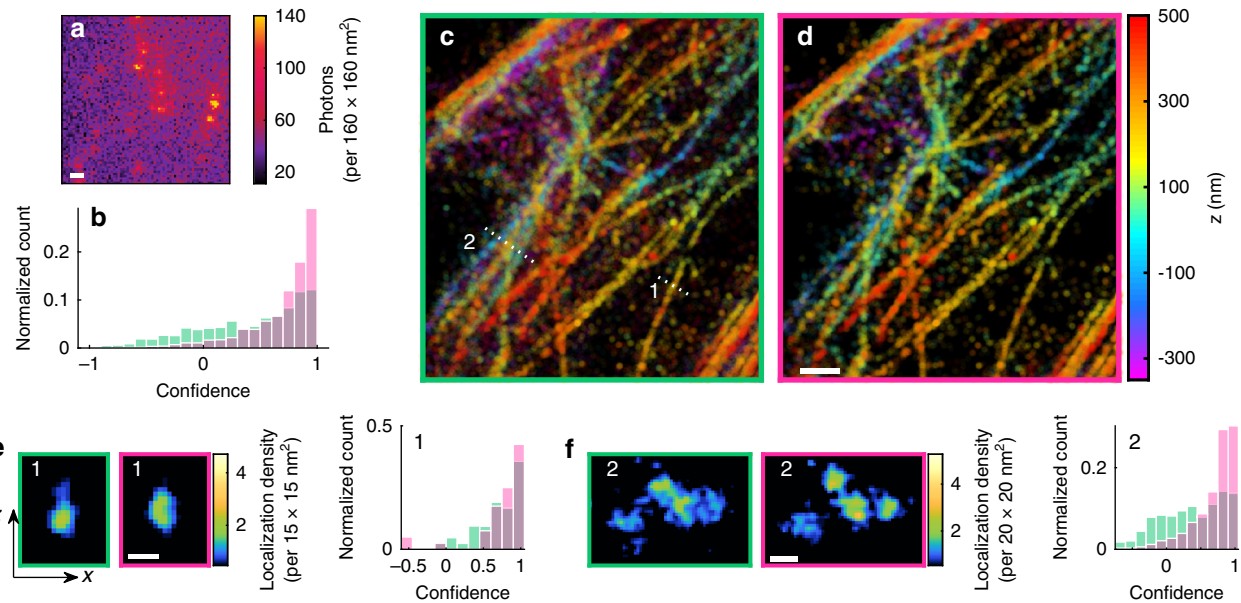

**Fig. 3 Comparison of 3D SMLM algorithms on experimental images of Alexa Fluor 647-labeled microtubules. a** Representative imaging frame of blinking SMs using the DHPSF. **b** Histogram of confidences of the Easy-DHPSF algorithm (green) and RoSE (red) corresponding to 500 randomly chosen frames from the 10,718-frame dataset. **c** 3D SMLM image of Easy-DHPSF localizations with apparent brightness greater than 1300 photons, color-coded as a function of $z$ position. **d** Similar to (**c**), but for RoSE localizations. **e** Transverse ($xz$) images and WIF distributions of localizations along the dotted line (1) in (**c**) (window width = 492 nm) corresponding to Easy-DHPSF (green) and RoSE (red). **f** Similar to (**e**), but for the dotted line (2) in (**c**) (window width = 726 nm). Color bars: **a** photons detected per $160 \times 160$ nm$^2$, **c**, **d** depth (nm), **e** localization density per $15 \times 15$ nm$^2$, **f** localization density per $20 \times 20$ nm$^2$. Scale bars: **a**, **d** 1 μm, **e** 100 nm, and **f** 200 nm.

quantify the performance of localization algorithms on individual localizations and subregions within SMLM reconstructions.

**Quantifying algorithmic robustness and molecular heterogeneity.** Next, we used WIF to characterize algorithmic performance for transient amyloid binding (TAB)[25,26] imaging of amyloid fibrils ("Methods"). Here, the relatively large shot noise in images of Nile red (<1000 photons per frame) tests the robustness of three distinct algorithms: TS with weighted-least squares (WLS) using a weighted Gaussian noise model; TS with maximum likelihood estimation (MLE) using a Poisson noise model; and RoSE, which uses a Poisson noise model but also is robust to image overlap.

Qualitative and quantitative differences are readily noticeable between reconstructed images, particularly where the fibrillar bundle unwinds (Fig. 4a–c, insets). We attribute the poor localization of WLS, exemplified by broadening of the fibrils (40 nm full-width at half-maximum [FWHM] of the well-resolved region within the dashed white box, Fig. 4a), to its lack of robustness to shot noise. By using instead a Poisson noise model, MLE recovers marginally thinner (39 nm FWHM) and better resolved fibrils, but struggles to resolve fibrils at the top end of the structure (Fig. 4b, e). This inefficiency is probably due to algorithmic failure on images containing overlapping molecules. In contrast, RoSE localizations have greater precision and accuracy (27 nm FWHM), thereby enabling the parallel unbundled filaments to be resolved (Fig. 4c, f). These perceived image qualities are reliably quantified via WIF. Indeed, RoSE localizations show the greatest confidence of the three algorithms with WIF$_{avg}$ = 0.78 while WLS shows a low WIF$_{avg}$ of 0.18, attesting to their excellent and poor recovery, respectively (Fig. 4g–i). Interestingly, we found that, in terms of FRC[10], RoSE has only 3% better resolution compared to MLE.

To further confirm that WIF is a reliable measure of accuracy at the single-molecule level, we filtered out all localizations with

confidence smaller than 0.5. Remarkably, filtered reconstructions from all three algorithms appear to resolve unbundled fibrils (Fig. 4j–l and Supplementary Figs. 25c, 26c).

In contrast, filtering based on estimated PSF width produces sub-optimal results. Notably, retaining MLE localizations within a strict width range $W_1 \in [90, 110$ nm$]$ improves filament resolvability at the cost of compromising sampling continuity (Supplementary Fig. 25a). For a slightly larger range, $W_2 \in [70, 130$ nm$]$, the filtering is ineffective and the fibrils are not well resolved (Supplementary Fig. 25b). Similarly, using estimated localization precision[27] as a filter, which is largely equivalent to using the estimated SM brightness, removes many useful localizations while also retaining bridging artifacts between individual fibers (Supplementary Figs. 26a, b).

A powerful feature of WIF is its ability to quantify an arbitrary discrepancy between a computational imaging model and SMLM measurements. This property is particularly useful since hidden physical parameters, which may be difficult to model accurately, can induce perturbations in the observed PSF. Therefore, we can use WIF to interrogate variations in the interactions of Nile red with amyloid fibrils that are encoded as subtle features within SMLM images. To demonstrate this capability, we analyzed TAB datasets using RoSE and calculated the WIFs of localizations with >400 detected photons (Fig. 5). Interestingly, WIF density plots reveal heterogeneous regions along both fibrils. Specifically, for segments of fibrils that are oriented away from the vertical axis, we see a larger fraction of localizations that have low confidence (<0.5) compared to regions that are vertically oriented (Fig. 5a, b). Quantitatively, the upper regions of two fibrils have 17% (Fig. 5c) and 37% (Fig. 5d) more localizations with confidence >0.8 compared to the bottom regions.

To examine the origin of this heterogeneity, we directly compare observed PSFs from high- and low-confidence regions. Curiously, PSFs in the bottom regions are slightly elongated along an axis parallel to the fibril itself, whereas PSFs from the top

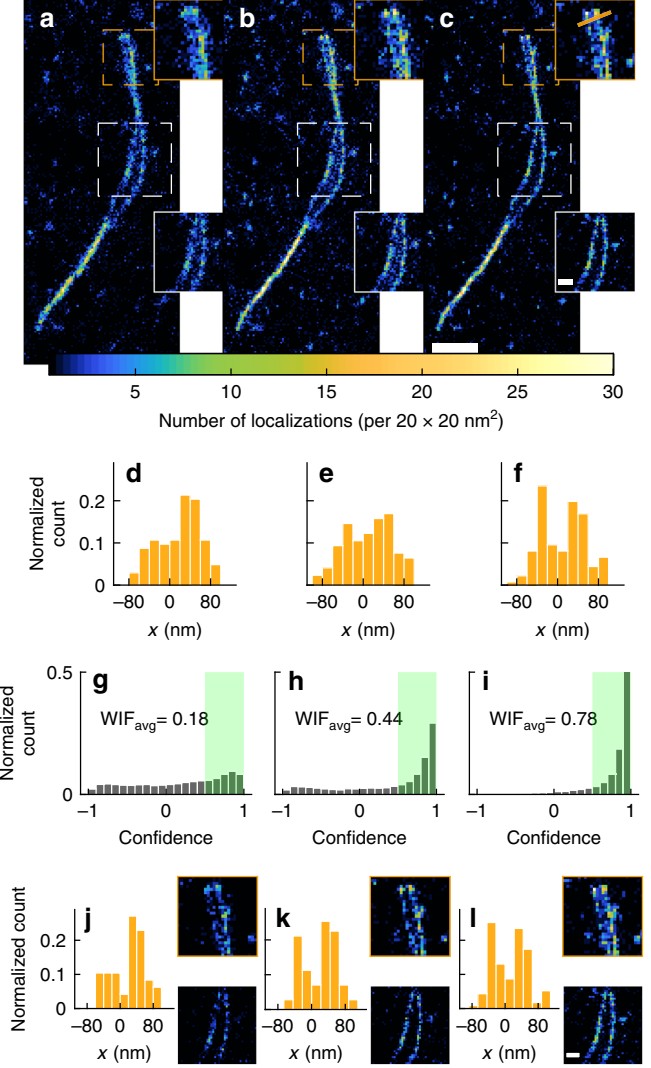

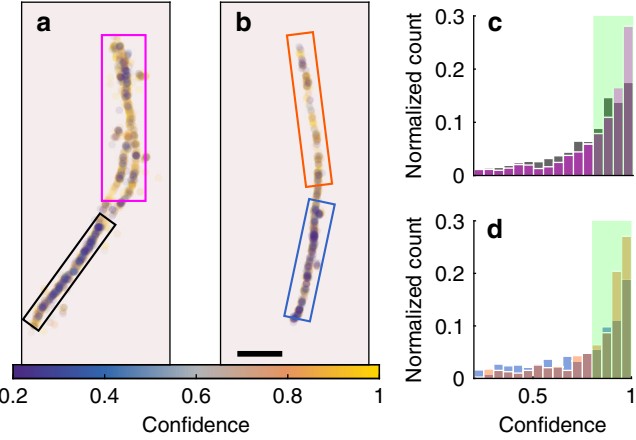

**Fig. 5 Heterogeneity in Nile red interactions with amyloid fibrils.**
**a**, **b** WIFs of bright localizations (>400 detected photons) detected by RoSE on two fibrils. **c**, **d** WIFs of localizations within corresponding boxed regions in **a** (upper magenta and lower black) and **b** (upper orange and lower blue), respectively. Green regions indicate localizations with WIF > 0.8, corresponding to 63% (**a**, magenta), 54% (**a**, black), 62% (**b**, orange), and 45% (**b**, blue) of the localizations. Color bar: confidence. Scale bar: 500 nm.

**Fig. 4 Quantifying algorithmic robustness and enhancing reconstruction accuracy in SMLM of amyloid fibrils.** Super-resolution image of twisted fibrils recovered by **a** TS-weighted least-squares (WLS), **b** TS maximum-liklihood estimation (MLE), and **c** RoSE. **d–f** Histograms of localizations projected onto the gold line in **c** (top inset) from **d** WLS, **e** MLE, and **f** RoSE. WIFs for **g** all WLS localizations in (**a**), **h** all MLE localizations in (**b**), and **i** all RoSE localizations in (**c**). Green regions denote localizations with confidence >0.5. **j–l** Histograms of localizations with confidence >0.5 projected onto the golden line in **c** (top inset) and corresponding filtered inset images for **j** WLS, **k** MLE, and **l** RoSE. Color bar: number of localizations per $20 \times 20$ nm$^2$. Scale bars: **c** 500 nm, **c** inset 150 nm, and **l** inset 150 nm.

regions better match our model (Supplementary Fig. 27). These features may be attributed to specific binding orientations of Nile red molecules to amyloid fibrils[26,28–30] in TAB imaging. We note that instrument aberrations may introduce confounding errors when uncovering the true source of heterogeneity. Therefore, when using WIF to detect the source of heterogeneity, e.g., alignment of molecules w.r.t. amyloid fibrils, it may be necessary to calibrate the PSF over the field-of-view.

## Discussion

WIF is a computational tool that utilizes mathematical models of the imaging system and measurement noise to measure the statistical confidence of each localization within an SMLM image. We used WIF to benchmark the accuracy of SMLM algorithms on a variety of simulated and experimental datasets. We also demonstrated WIF for analyzing how sample non-idealities affect reconstruction accuracy. Intuitively, low signal to noise ratios make the detection of minor model mismatches, such as defocus, comparatively difficult (Supplementary Fig. 6). While WIF has excellent sensitivity for detecting overlapping molecules (Fig. 1d), dipole-like emission patterns (Fig. 1e and Supplementary Figs. 7, 8), and sub-optimal localization algorithms (Fig. 3 and Supplementary Fig. 15), WIF cannot explain the source of low confidence values that cause localization inaccuracies or heterogeneities; rather, it detects and quantifies these effects without needing knowledge of a specific mismatch to search for. Nonetheless, the geometry of the transport trajectories themselves can provide insight into the specific mismatch observed in the data (Supplementary Fig. 11).

WIF exhibits several advantages over existing methods for quantifying reconstruction accuracy in experimental SMLM. First, WIF does not require labeled training data to judge the trustworthiness of predetermined image features; a model of the imaging system PSF and statistical noise suffices (see Supplementary Fig. 29 and Supplementary Note 3 for an example of how mismatch in the noise model leads to inconsistent WIF scores), which can be obtained through calibration techniques[16,31–33]. Second, it does not need ground-truth knowledge of SM positions, which would be prohibitive in most SMLM applications. Third, it obviates the need to align SMLM images to a secondary imaging modality for comparison and is therefore more robust than such approaches. More fundamentally, WIF exploits a unique property of SMLM compared to other non-SM super-resolution optical methodologies (e.g., structured illumination, RESOLFT, and STED); imaging the entirety (peak and spatial decay) of each SM PSF synergistically creates well-behaved gradient flows along the likelihood surface that are used in computing WIF. Finally, computing mismatches in image space (e.g., PSF width in Fig. 1c–f) is insensitive to molecular overlap, defocus, and dipole emission artifacts without assuming strong statistical priors on the spatial distribution of molecules or a simplified PSF[34].

WIF can be used for online tuning of parameters (e.g., activation density and imaging buffer conditions) during an experiment to maximize imaging accuracy and resolution. It also offers a reliable means to detect otherwise hidden properties of the sample of interest, such as molecular orientation shown here, allowing for the discovery of new biophysical and biochemical phenomena at the nanoscale. While a majority of neural network training methods in SMLM utilize simulated data[35] or experimental data assuming a perfectly matched model[36], the discriminative power of WIF may enable these networks to be trained robustly on experimental data in the presence of mismatches stemming from hidden parameters[37].

WIF represents an advance in statistical quantification in image science[38], where the reliability of each quantitative feature within a scientific image can now be evaluated. The benefits of integrating WIF into downstream analysis[39] (e.g., SM clustering, counting, and co-localization) and even in other imaging modalities (e.g., spectroscopy, astronomical imaging, positron emission tomography, and computed tomography) are exciting opportunities yet to be explored.

## Methods

**Definitions and notations**. In this section we define terms used in deriving and computing WIF.

*Notations*. We represent vectorial quantities in bold. For a vector $\mathbf{v}$, we denote its Euclidean norm by $\|\mathbf{v}\|$. We use $\delta(\mathbf{v})$ to represent the Dirac delta function, which is zero everywhere except at $\mathbf{v} = \mathbf{0}$. The inner product of two vectors $\mathbf{v}_1$ and $\mathbf{v}_2$ is denoted by $\mathbf{v}_1 \cdot \mathbf{v}_2$. Further, we denote $[\mathcal{N}]$ as the set of integers $\{1, \dots, \mathcal{N}\}$. Finally, we use $\hat{x}$ to represent an estimate of a deterministic quantity $x$.

*A set of localizations*. We represent a set of localizations or a set of source molecules as

$$\hat{\mathcal{M}} = \sum_{i=1}^{\hat{N}} \hat{s}_i \delta(\mathbf{r} - \hat{\mathbf{r}}_i), \tag{1}$$

where $\hat{s}_i \geq 0$ and $\hat{\mathbf{r}}_i \in \mathbb{R}^2$ denote the $i$th molecules' estimated brightness (in photons) and position, respectively. Note that, throughout this paper, what we mean by brightness is the expected number of photons emitted by a molecule during a camera frame (see ref. [40] for background). We denote the mass of $\hat{\mathcal{M}}$, i.e., the sum of the brightnesses of all molecules in $\hat{\mathcal{M}}$, by $\hat{S}$. Further, $\hat{N}$ represents the number of molecules in $\hat{\mathcal{M}}$. We represent the collection of all valid $\hat{\mathcal{M}}$ by M.

*Negative Poisson log likelihood*. Consider a set of $\hat{N}$ molecules given by $\hat{\mathcal{M}}$. The resulting intensity $\mu_j$, that is, the expected number of photons detected on a camera, for each pixel $j \in \{1, \dots, m\}$ can be written as

$$\mu_j = \sum_{i=1}^{\hat{N}} \{\hat{s}_i q_j(\hat{\mathbf{r}}_i)\} + b_j, \quad q_j(\hat{\mathbf{r}}_i) = \int_{C_j} q(\mathbf{u} - \hat{\mathbf{r}}_i) d\mathbf{u}, \tag{2}$$

where $q_j(\hat{\mathbf{r}}_i)$ represents the integrated value of the PSF $q$ (for the $i$th molecule) over the $j$th pixel area ($C_j$), and $b_j$ denotes the expected number of background photons at the $j$th pixel.

If we denote $\mathbf{g} \in \mathbb{R}^m$ as $m$ pixels of photon counts captured by a camera, the negative Poisson log likelihood $\mathcal{L}$ is then given by

$$\mathcal{L}(\hat{\mathcal{M}}) = \sum_{j=1}^{m} \{\mu_j - g_j \log(\mu_j)\}, \tag{3}$$

where we have discarded terms that do not depend on $\hat{\mathcal{M}}$. If $q$ is the true PSF, we call $\mathcal{L}$ the true negative log likelihood, while conversely, if an estimated or candidate model PSF $q$ is used, then, we refer to $\mathcal{L}$ as the negative log likelihood of the model. We note that the Poisson noise model considered here can be extended to account for pixel-dependent readout (Gaussian) noise (Supplementary Note 3)[41].

*Grid points in object space*. We consider a set of $\mathcal{N}$ Cartesian grid points represented by $\mathcal{G} = \{\mathbf{r}_{\mathcal{G}_i}\}_{i=1:\mathcal{N}}$ for which the distance between any two adjacent grid points is given by $2\rho$. In this way, a set of localizations can be uniquely represented via a discrete grid $\mathcal{G}$:

$$\hat{\mathcal{M}} = \sum_{i=1}^{\hat{N}} \hat{s}_{[i]} \delta(\mathbf{r} - \hat{\mathbf{r}}_{[i]}), \tag{4}$$

where $[i]$ represents a grid point index in $[\mathcal{N}]$, $\hat{\mathbf{r}}_{[i]} = \hat{\mathbf{r}}_{\mathcal{G}_{[i]}} + \Delta \hat{\mathbf{r}}_{[i]}$, $\hat{\mathbf{r}}_{\mathcal{G}_{[i]}}$ is the closest grid point to the $i$th molecule, and $\Delta \hat{\mathbf{r}}_{[i]}$ denotes a position offset (Supplementary Note 3).

*Local perturbation*. We perturb a set of localizations $\hat{\mathcal{M}}$ by introducing a small distortion in the positions and brightnesses of the molecules in $\hat{\mathcal{M}}$ to produce another set of localizations $\mathcal{M}_0$:

$$\mathcal{M}_0 = \sum_{i=1}^{\hat{N}} \sum_{j=1}^{8} \hat{s}_{[i,j]} \delta(\mathbf{r} - \hat{\mathbf{r}}_{[i,j]}), \tag{5}$$

where $\sum_{j=1}^{8} \hat{s}_{[i,j]} = \hat{s}_{[i]}$ and $\hat{\mathbf{r}}_{[i,j]}$ is one of the eight neighboring grid points of $\hat{\mathbf{r}}_{\mathcal{G}_{[i]}}$ (Fig. 1b). We denote $\Delta \hat{\mathbf{u}}_{[i,j]} \triangleq (\hat{\mathbf{r}}_{[i]} - \hat{\mathbf{r}}_{[i,j]})/\|\hat{\mathbf{r}}_{[i]} - \hat{\mathbf{r}}_{[i,j]}\|$ as a unit perturbation vector (Fig. 1b).

*Wasserstein distance*. We define the Wasserstein distance $\mathbb{W}_2$ between two sets of localizations[42,43], $\mathcal{M}_1 \in$ M and $\mathcal{M}_2 \in$ M, with equal masses as the minimum cost of transporting one to the other among all valid transportation plans $\Pi$:

$$\mathbb{W}_2(\mathcal{M}_1, \mathcal{M}_2) = \sqrt{\min_{\pi \in \Pi} \left( \sum_{i=1}^{\hat{N}_1} \sum_{j=1}^{\hat{N}_2} \|\hat{\mathbf{r}}_i - \hat{\mathbf{r}}_j\|_2^2 \pi(\hat{\mathbf{r}}_i, \hat{\mathbf{r}}_j) \right)}, \tag{6}$$

where $\pi(\hat{\mathbf{r}}_i, \hat{\mathbf{r}}_j)$ is the portion of photons from the molecule at position $\hat{\mathbf{r}}_i$ in $\mathcal{M}_1$ that is transported to the position $\hat{\mathbf{r}}_j$ in $\mathcal{M}_2$.

**Derivation of WIF**. WIF is derived based on the mathematical notion that accurate localizations are global minima of the true negative log likelihood. Therefore, any small, arbitrary perturbation of the true localizations will absolutely increase the true negative log likelihood (Eq. (3)). In contrast, for inaccurate localizations, we can find a local perturbation such that by transporting the localizations along some perturbation trajectory, the true negative log likelihood decreases (Eq. (3)). In the following subsections, we make these observations precise.

Given a set of input localizations, $\hat{\mathcal{M}}$, we aim to find a local perturbation that minimizes the negative log likelihood of our model $\mathcal{L}$:

$$\mathcal{M}_1 = \underset{\mathcal{M} \in \text{M}: \mathbb{W}_2^2(\hat{\mathcal{M}}, \mathcal{M}) \leq \zeta'}{\arg \min} \mathcal{L}(\mathcal{M}), \tag{7}$$

where $\zeta'$ signifies the degree of uncertainty in $\hat{\mathcal{M}}$ expressed as the square of the radius of the Wasserstein ball around $\hat{\mathcal{M}}$. For example, if $\zeta' = 0$, signifying absolute certainty in the input localizations, then we get $\mathcal{M}_1 = \hat{\mathcal{M}}$.

Alternatively, and perhaps more revealing, we can express Eq. (7) by shifting the center of the uncertainty ball to $\mathcal{M}_0$ (Eq. (5)):

$$\mathcal{M}_1 = \underset{\mathcal{M} \in \text{M}: \mathbb{W}_2^2(\mathcal{M}_0, \mathcal{M}) \leq \epsilon'}{\arg \min} \mathcal{L}(\mathcal{M}), \tag{8}$$

where $\epsilon'$ is related to $\zeta'$ in Eq. (7). The solution to Eq. (8) can be expressed as

$$\mathcal{M}_1 = \sum_{i=1}^{\hat{N}} \sum_{j=1}^{8} \hat{s}_{[i,j]} \delta(\mathbf{r} - \tilde{\mathbf{r}}_{[i,j]}). \tag{9}$$

The set $\mathcal{M}_1$ characterizes the transport trajectories $\Delta \tilde{\mathbf{r}}_{[i,j]} \triangleq \tilde{\mathbf{r}}_{[i,j]} - \hat{\mathbf{r}}_{[i,j]}$ for each source molecule in $\hat{\mathcal{M}}$ (Fig. 1b), where $\hat{\mathbf{r}}_{[i,j]}$ are the set of perturbed molecule positions (Eq. (5)) and $\tilde{\mathbf{r}}_{[i,j]}$ are the molecule positions in $\mathcal{M}_1$. These trajectories allow us to measure the stability in the position $\hat{\mathbf{r}}_i$ of a molecule in $\hat{\mathcal{M}}$ (Fig. 1b): if $\hat{\mathbf{r}}_i$ is perfectly stable, then $\Delta \tilde{\mathbf{r}}_{[i,j]}$ should point toward $\hat{\mathbf{r}}_i$ for all $j \in \{1, \dots, 8\}$. Therefore, we define WIF of a source molecule as the portion of photon flux that returns toward it after a local perturbation (Eq. (5)) and regularized transport (Eq. (8)):

$$\text{WIF} \triangleq \frac{\sum_{j=1}^{8} \hat{s}_{[i,j]} \Delta \tilde{\mathbf{r}}_{[i,j]} \cdot \Delta \hat{\mathbf{u}}_{[i,j]}}{\sum_{j=1}^{8} \hat{s}_{[i,j]} \|\Delta \tilde{\mathbf{r}}_{[i,j]}\|}. \tag{10}$$

WIF takes values in $[-1, 1]$ where 1 represents a source molecule with the highest confidence. We justify rigorously the definition of WIF presented in Eq. (10) using the theory of Wasserstein gradient flows (Supplementary Note 1)[18].

**Computing WIF**. Solving for $\mathcal{M}_1$ in Eq. (8) is challenging, which involves an inner optimization of the Wasserstein distance. To obtain an efficient algorithm to compute WIF, we first consider an equivalent form of Eq. (8) using Lagrangian relaxation as:

$$\mathcal{M}_1 = \arg \min_{\mathcal{M} \in \text{M}} \{\mathbb{W}_2^2(\mathcal{M}_0, \mathcal{M}) + \epsilon \mathcal{L}(\mathcal{M})\}, \tag{11}$$

where $\epsilon$ is related to $\epsilon'$ in Eq. (8). Next, by approximating the continuous position of source molecules using a set of grid points and position offsets, we can find a tractable upper bound for Eq. (8) that can be solved efficiently via accelerated proximal-gradient methods (Supplementary Note 3). In particular, we compute

$\mathcal{M}_1$ according to the following regularized, negative log likelihood minimization:

$$\mathcal{M}_1 = \arg\min_{\mathcal{M}\in\mathcal{C}\cap\mathbf{M}} \{\nu\mathcal{R}(\mathcal{M}) + \mathcal{L}(\mathcal{M})\}, \tag{12}$$

where $\nu > 0$ is inversely related to $\epsilon$, $\mathcal{R}(\mathcal{M}) = \sum_{i=1}^{N} \sqrt{s_i^2 + s_i^2\|\Delta\mathbf{r}_i\|_2^2}$ is a group-sparsity norm, and $\mathcal{C}$ is a constraint set that limits $\Delta\mathbf{r}_i$ to be within its neighboring grid points (see Supplementary Note 3 and Supplementary Table 2 for details).

**Extension to 3D SMLM.** A natural extension of WIF to 3D imaging involves locally perturbing an estimated molecule within a small volume. Such a strategy complicates the computation of WIF as it requires a distinct PSF model for each molecule in the imaging volume. With this complexity in mind, we consider a variant of WIF in 3D that lends itself to an efficient algorithm, which is identical to that of WIF in 2D. Specifically, we capitalize on the observation that an accurate localization in 3D should not only be stable w.r.t. a volumetric perturbation but also w.r.t. an in-plane ($xy$) perturbation. Therefore, for any estimated molecule, we consider a 2D perturbation similar to Eq. (5) in which each perturbed source molecule maintains the same axial position of the original estimated molecule (Supplementary Note 3).

**Processing experimental data.** Prior to analyzing experimental data, we first estimated a pixel amplitude-offset map by averaging 100 camera frames with the shutter closed. The offset map was subtracted from the raw camera images (pixels with values smaller or equal than zero were set to $10^{-3}$). Next, we converted the offset-subtracted images to photon counts based on the conversion gain of each camera (assuming uniformity across the field-of-view, see below). These images were then used for SM localization, WIF calculation, and estimating fluorescence background (Supplementary Note 5).

**Super-resolution imaging of labeled microtubules.** The microtubules of BSC-1 cells were immunolabeled with Alexa Fluor 647 (Invitrogen) and imaged under blinking conditions[44] with glucose oxidase/catalase and mM concentrations of mercaptoethylamine (MEA) as in ref. [45]. The sample was imaged using an Olympus IX71 epifluorescence microscope equipped with a 100× 1.4 NA oil-immersion objective lens (Olympus UPlan-SApo 100×/1.40). Fluorophores were excited using a 641-nm laser source (Coherent Cube, peak intensity ~10 kW cm$^{-2}$). Fluorescence from the microscope was filtered using a dichroic beamsplitter (Semrock, Di01-R635) and bandpass filter (Omega, 3RD650-710) and separated into two orthogonally polarized detection channels. Both polarized channels reflect off of a phase-only spatial light modulator (Boulder Nonlinear Systems, SLM) placed in the Fourier plane before being imaged onto a camera; a flat phase pattern was used for the 2D SMLM experiments (Fig. 2), while the double-helix phase mask was used for the 3D data as in ref. [45] (Fig. 3). Fluorescence photons were captured using an electron-multiplying (EM) CCD camera (Andor iXon+ DU897-E) at an EM gain setting of 300 with a pixel size of $160 \times 160$ nm$^2$ in object space and a conversion gain of 0.13 ADU photon$^{-1}$. Only one polarization channel was analyzed in this work. For the 284k localizations shown in Fig. 2b, 2287 photons were detected on average with a background of 76 photons per pixel. For the 48,445 localizations shown in Fig. 3d, 2238 photons were detected on average with a background of 89 photons per pixel. SMLM images in Fig. 3 were rendered by plotting each localization as a symmetric 2D Gaussian function with a standard deviation of (Fig. 3e) 15 nm and (Fig. 3f) 20 nm.

**Transient Amyloid Binding imaging.** The 42 amino-acid residue amyloid-beta peptide (A$\beta$42) was synthesized and purified by Dr. James I. Elliott (ERI Amyloid Laboratory, Oxford, CT) and dissolved in hexafluoro-2-propanol (HFIP) and sonicated at room temperature for 1 h. After flash freezing in liquid nitrogen, HFIP was removed by lyophilization and stored at −20 °C. To further purify the protein, the lyophilized A$\beta$42 was dissolved in 10 mM NaOH, sonicated for 25 min in a cold water bath and filtered first through a 0.22 μm and then through a 30 kDa centrifugal membrane filter (Millipore Sigma, UFC30GV and UFC5030) as described previously[25]. To prepare fibrils, we incubated 10 μM monomeric A$\beta$42 in phosphate-buffered saline (PBS, 150 mM NaCl, 50 mM Na$_3$PO$_4$, pH 7.4) at 37 °C with 200 rpm shaking for 20–42 h. The aggregated structures were adsorbed to a ozone-cleaned cell culture chamber (Lab Tek, No. 1.5H, 170 ± 5 μm thickness) for 1 h followed by a rinse using PBS. A PBS solution (200 μL) containing 50 nM Nile red (Fisher Scientific, AC415711000) was placed into the amyloid-adsorbed chambers for transient amyloid binding.

Blinking Nile red molecules on fibrils were imaged using a home-built epifluorescence microscope equipped with a 100× 1.4 NA oil-immersion objective lens (Olympus, UPlan-SApo 100×/1.40). The samples were excited using a 561-nm laser source (Coherent Sapphire, peak intensity ~0.88 kW cm$^{-2}$). Fluorescence was filtered by a dichroic beamsplitter (Semrock, Di03-R488/561) and a bandpass filter (Semrock, FF01-523/610) and separated into two orthogonally polarized detection channels by a polarizing beamsplitter cube (Meadowlark Optics). Both channels were captured by a scientific CMOS camera (Hamamatsu, C11440-22CU) with a pixel size of $58.5 \times 58.5$ nm$^2$ in object space and a conversion gain of 0.49 ADU photon$^{-1}$. Only one of the channels was analyzed in this work. For the 12k localizations shown in Fig. 4c, 390 photons were detected on average with a background of five photons per pixel. For the 931 localizations shown in Fig. 5b, 785 photons were detected on average with a background of 2.4 photons per pixel.

**Synthetic data.** We generated images of molecules via a vectorial image-formation model[19], assuming unpolarized ideal PSFs. Briefly, a molecule is modeled as a dipole rotating uniformly within a cone with a half-angle $\alpha$. A rotationally fixed dipole corresponds to $\alpha = 0$, while $\alpha = 90°$ represents an isotropic molecule. Molecular blinking trajectories were simulated using a two-state Markov chain[7]. We used a wavelength of 637 nm, NA = 1.4, and spatially uniform background. We simulated a camera with $58.5 \times 58.5$ nm$^2$ square pixels in object space.

**Jaccard index.** Following ref. [8], given a set of ground-truth positions and corresponding localizations, we first match these points by solving a bipartite graph-matching problem of minimizing the sum of distances between the two elements of a pair. We say that a pairing is successful if the distance between the corresponding two elements is smaller than twice the full width at half maximum (FWHM) of the localization precision $\sigma$, which is calculated using the theoretical Cramér−Rao bound ($\sigma = 3.4$ nm with 2000 photons detected). The elements that are paired with a ground-truth position are counted as true positives (TPs) and those without a pair are counted as false positives (FPs). Finally, the ground-truth molecules without a match are counted as false negatives (FNs). The Jaccard index is calculated as TP/(TP + FP + FN).

**PSF modeling for computing Wasserstein-induced flux.** For simulation studies, we used an ideal, unpolarized standard PSF resulting from an isotropic emitter (Fig. 1 and Supplementary Figs. 3, 4, 6–8, 10–16, 18, 29), while for experimental data (Figs. 2, 4, 5 and Supplementary Figs. 19, 25–27), we used a linearly-polarized PSF, also resulting from an isotropic emitter (see Supplementary Table 2 for details).

In addition to the ideal PSFs modeled above, we needed to calibrate the aberrations present in the PSF used for microtubule imaging (Fig. 2). We modeled the microscope pupil function $P$ as

$$P(u, v) = \exp\left(j\sum_{i=3}^{l}\{a_i Z_i(u, v)\}\right) \cdot P_0(u, v), \tag{13}$$

where $(u, v)$ are microscope's pupil coordinates; $Z_i$ and $a_i$ represent the $i$th Zernike basis function and its corresponding coefficient; and $P_0$ denotes the pupil function of the uncalibrated model. We used 33 Zernike modes corresponding to $l = 35$.

Using RoSE, we localized well-isolated molecules over a large FOV corresponding to Fig. 2. Next, for each localization, we extracted a raw image of size $11 \times 11$ pixels with the localized molecule at its center. We excluded molecules with brightnesses less than 3000 photons or with positions away from the origin by more than one pixel. Next, we randomly selected 600 of these images to estimate the Zernike coefficients, i.e., $\{a_1, ..., a_l\}$, as described previously[46]. The calibrated PSF (Supplementary Fig. 19) is then computed based on the recovered pupil $P$.

Previous works on 3D PSF modeling applied robust averaging of finely sampled axial scans of many beads by aligning them using polynomial interpolation[47]. Here, we further robustify these methods by employing tools from OT[43]. Specifically, we use displacement interpolation to obtain PSFs with a $z$ spacing of 10 nm from a reference bead scan taken at 40-nm axial intervals. We next use additional beads to augment this model from the reference bead. We first estimate the $z$ position of a bead at top of the stack. Next, we use OT to obtain PSFs at model $z$ planes. Next, we register the two sets of PSF scans laterally using cubic interpolation. We repeat this same process for other non-reference beads. Finally, we use B-splines to smooth the 3D PSF (used for 3D microtubule imaging, Fig. 3) and obtain PSF gradients in 3D (Supplementary Fig. 20).

**Reporting summary.** Further information on research design is available in the Nature Research Reporting Summary linked to this article.

## Data availability

Data that support the findings of this study is available at https://osf.io/d72zv/?view_only=f7b47b8b542246d1a326d8b8a8c3a60f.

## Code availability

Related codes that support the findings of this study are available at https://osf.io/d72zv/?view_only=f7b47b8b542246d1a326d8b8a8c3a60f.

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

## Acknowledgements

We thank W.E. Moerner and Steffen J. Sahl for contributing the SMLM microtubule dataset. We also thank Jin Lu, Oumeng Zhang, Tingting Wu, and Zhenqi Lu for helpful discussion. Research reported in this publication was supported by the National Science Foundation under grant number ECCS-1653777 and by the National Institute of General Medical Sciences of the National Institutes of Health under grant number R35GM124858 to M.D.L.

## Author contributions

H.M. and M.D.L. designed the research; H.M. designed and implemented analysis algorithms and performed data analysis; T.D. designed and performed Transient Amyloid Binding (TAB) experiments. M.D.L. and A.N. supervised the research; all authors wrote the paper.

## Competing interests

The authors declare that U.S. Provisional Patent Application 62/880,495 has been filed covering the methodology in this manuscript.
