## [Peer Review File · Nature Communications]

Reviewers' Comments:

Reviewer #1:

Remarks to the Author:

Mazidi, et al. propose a computational method, Wasserstein-induced flux (WIF), to estimate the confidence of each localization without prior knowledge of molecules. The authors evaluated the WIF under several experimental conditions, including 2D, 3D using double-helix, and different SNRs and brightness. They confirmed the robustness of WIF using different samples and used WIF for heterogeneity identification.

The manuscript is well written and presented with sufficient investigations. This reviewer recommends publishing this manuscript after some minor revisions to address concerns below.

1. What is the definition of SNR in the PSF model? Please take a look at this Ref. – Long et al, Optics Express 20, 17741 (2012) and define the SNR at the beginning of the manuscript.

2. As the author pointed out that WIF does not explain the true source of errors, which remains quite confusing to this reviewer. For example, the author mentioned that the origin of the heterogeneity of fibrils in Fig. 5 may come from binding orientations. Although I agree that it is likely to be the case, exaggeratively, it could be originated from any source, field-dependent aberrations, the system imperfection, or interactions between molecules and their local environments. Knowing discrepancy itself may not be sufficient in actual studies of measuring heterogeneity. It would be more convincing if the authors could provide discussion or evidence to identify the nature of these errors.

3. In precision estimation, noise sources are modeled as Poisson and Gaussian distributions. Pixelation error (Not pixel-dependent readout noise) in the Ref. – Thompson et al, Biophysical Journal 82, 2775 (2002), also could impact WTF values?

4. The authors assessed the WIF performance considering defocus and rotational mobility and observed that confidence is relatively sensitive to defocusing, as shown in Fig. S6, within a range of 0-300 nm. Double-helix (DH) 3D method is well known by its relatively large depth range over +/- 1 micron. Thus, according to the given information, WIF may not get full benefits from the DH 3D method. Whether this trend can be generalized to any other common 3D PSF imaging approaches, including astigmatism, biplane, or any other defocusing PSF model? Then, the results and analyses described in the Supplementary Note 6 and Fig. S6 may be different depending on the different approaches? If so, what is the best 3D imaging approach suitable for WIF?

Reviewer #3:

Remarks to the Author:

The authors answered in a very complete manner to my requests and questions. The manuscript has been improved from the reviewers' comments.

The authors provide a metric for each molecule without the need of ground-truth positions (as opposed to Jaccard Index, etc.).

This manuscript offers a tool which was missing for real SMLM data.

One comment:

Their metric does not attempt to explain the source of low confidence, but the authors proposed diverse flavours of WIF (geometry of transport or others) to provide some insights.

Mainly, this behaviour is correct if the likelihood used by WIF is accurate. Otherwise, as shown by the authors, WIF can yield an opposite scenario if the noise model is inaccurate.

Another aspect is that some training data are probably needed to calibrate the hyper-parameters for new applications, although the authors suggest that their choice of hyper-parameters is fairly robust.

These aspects suggest that some expertise and carefulness are required for its usage and interpretation, but this is also the case for FRC/FSC when applying on real SMLM data.

Globally, the revised paper is clearer and more honest regarding the cautions to take, and the authors assess a large panel of scenarios. The additional experiments also show several behaviours of WIF (e.g., wrt regularizer strength) and how to take advantage of them. These will help the readers to better understand the proposed tool.

In the spirit of open science, I strongly encourage the authors to publish a usable / user-friendly software. The documentation should also state the limitations and/or cautions the user should take. That would only help for its dissemination and usage.

In conclusion, I support the publication of the submitted paper.

Point-by-point response to reviewer comments

“Quantifying accuracy and heterogeneity in single-molecule super-resolution microscopy”

We thank the reviewers for their constructive comments, which have spurred us to improve the clarity and quality of the manuscript. Below, we provide our point-by-point responses to the reviewers’ comments. The reviewers’ comments and remarks are presented in *italic* and black; our response in green; and any additions to the manuscript in magenta. For editorial and reviewers’ convenience, we also include clean and “tracked changes” versions of the main text and SI with our submission.

Reviewer 1

Mazidi, et al. propose a computational method, Wasserstein-induced flux (WIF), to estimate the confidence of each localization without prior knowledge of molecules. The authors evaluated the WIF under several experimental conditions, including 2D, 3D using double-helix, and different SNRs and brightness. They confirmed the robustness of WIF using different samples and used WIF for heterogeneity identification.

The manuscript is well written and presented with sufficient investigations. This reviewer recommends publishing this manuscript after some minor revisions to address concerns below.

1. *What is the definition of SNR in the PSF model? Please take a look at this Ref. – Long et al, Optics Express 20, 17741 (2012) and define the SNR at the beginning of the manuscript.*

We have now added a definition of SNR when we first introduce it, consistent with the work of Long et al. in its simplest form:

Here, we define (peak) SNR as the ratio of the number of photons (s_{sig}) in the brightest pixel of a PSF to the square root of the sum of s_{sig} and the detected background photons in that pixel [1].

2. *As the author pointed out that WIF does not explain the true source of errors, which remains quite confusing to this reviewer. For example, the author mentioned that the origin of the heterogeneity of fibrils in Fig. 5 may come from binding orientations. Although I agree that it is likely to be the case, exaggeratively, it could be originated from any source, field-dependent aberrations, the system imperfection, or interactions between molecules and their local environments. Knowing discrepancy itself may not be sufficient in actual studies of measuring heterogeneity. It would be more convincing if the authors could provide discussion or evidence to identify the nature of these errors.*

This certainly is a valid and important point raised by the reviewer. For studies of measuring and identifying the source of heterogeneity, care must be taken to calibrate the PSF and localization algorithm. Such calibrations usually remove instrument artifacts and enable true variations arising from the sample to be revealed [2, 3].

We stress that we did not design WIF to reveal the true source of errors; rather it provides a consistent and sensitive metric to quantify how well a certain imaging model explains observations. That is, existing metrics like PSF width, single-molecule brightness, and general statistical tests like the reduced chi-square statistic are insufficient for detecting errors that the reviewer mentioned, i.e., field-dependent aberrations, system imperfections, etc. In order for the scientist to detect heterogeneity and pinpoint its source through various control experiments, they must have a sensitive, reliable tool. Therefore, our manuscript proposes that WIF is such a tool for detecting and aiding the scientist to identify the most likely cause of heterogeneity.

To provide further evidence for the true source of heterogeneity observed in images of Nile red molecules, we have cited our recent work [3] on single-molecule orientation localization microscopy in which we simultaneously obtain super-resolved images of amyloid fibrils as well as the alignment of Nile red molecules w.r.t. the fibril “backbones.” Further, to clarify to the reader the potential challenges in detecting heterogeneity in SMLM, we have added the following sentences in the main text:

We note that instrument aberrations may introduce confounding errors when uncovering the true source of heterogeneity. Therefore, when using WIF to detect the source of heterogeneity, e.g., alignment of molecules w.r.t. amyloid fibrils, it may be necessary to calibrate the PSF over the field-of-view.

3. *In precision estimation, noise sources are modeled as Poisson and Gaussian distributions. Pixelation error (Not pixel-dependent readout noise) in the Ref. – Thompson et al, Biophysical Journal 82, 2775 (2002), also could impact WTF [sic] values?*

We thank the curious reviewer for this remark. As mentioned in Ref. [4], pixelation affects how the optical PSF is sampled by the camera, and therefore the achievable localization precision. The quantitative effect on localization precision depends on the detected number of photons from the emitter and the average background level. In the limiting case where background is zero, enlarging the background deteriorates (quadratically) the achievable localization precision. In addition, increased background intuitively causes the likelihood landscape around the true parameter to become less sharp (or have a smaller curvature). Therefore, we expect that the perturbed sources in the WIF computation to exhibit more random movements (due to shot noise) that deviate from converging to the true localization. These random displacements in turn reduce the average WIF.

To quantitatively illustrate the effect of pixelation on WIF, we simulated images of a molecule for two different camera pixel sizes: 58.5 nm and 160 nm. We localized the molecule in each image and computed its WIF using identical PSF models. Ideally, WIF values should be minimally affected by pixelation, with WIF medians close to 1 in both cases. However, we observe that the estimated WIFs for the larger pixel size (160 nm) are lower on average (Fig. 1).

This effect stems from our current implementation of WIF, which uses first order approximations of the PSF along the x and y axes. Therefore, a larger pixel size introduces larger residual errors, which adversely impact the estimated WIFs. For instance, residual or higher order errors could accumulate within a larger pixel, thereby exacerbating the first-order model mismatch. The first-order approximation was enacted for computational expediency, and future versions of WIF can use higher-order approximations when appropriate.

We have added the following sentences to the supplementary note 3D:

We also investigated the effect of first-order approximation errors for a large camera pixel size (Fig. S3b). A larger camera pixel size (160 nm, median WIF of 0.84) introduces larger residual errors as compared to a smaller camera pixel size (58.5 nm, median WIF of 0.94), which adversely impact the estimated WIFs.

Figure 1: Effect of camera pixelation on WIF. For each camera pixel size, 58.5 nm and 160 nm, we simulated 200 independent images of a molecule located at the origin. For both cases, we used an expected brightness of 2000 photons. For pixel size of 58.5 nm, we used an average, uniform background of 20 photons per pixel, while for 160 nm pixel size, we used a uniform background of 54.7 photons per pixel, which ensures that the background level is appropriately scaled with the camera pixel size. To compute WIF, we used the same standard PSF model that we used to localize these molecules, and the grid distances for both pixel sizes were roughly the same (grid distances of 30 nm and 40 nm for 58.5 nm and 160 nm pixel sizes, respectively). We used a regularizer value of 0.1 for both cases. We note that the achievable localization precision (Ref. [4]) of x and y for both cases are virtually the same (8.02 nm versus 8.08 nm), indicating that the difference in the WIF distributions is due to first-order approximation errors. Median WIFs: (58.5 nm pixel size) 0.94 and (160 nm pixel size) 0.84.

4. The authors assessed the WIF performance considering defocus and rotational mobility and observed that confidence is relatively sensitive to defocusing, as shown in Fig. S6, within a range of 0-300 nm. Double-helix (DH) 3D method is well known by its relatively large depth range over +/- 1 micron. Thus, according to the given information, WIF may not get full benefits from the DH 3D method. Whether this trend can be generalized to any other common 3D PSF imaging approaches, including astigmatism, biplane, or any other defocusing PSF model?

We believe that there may be a misinterpretation of the data; we apologize for the confusion. WIF is sensitive to model mismatch, which can occur for the standard 2D PSF or any 3D PSF. Fig. S6 illustrates that when we use WIF to compute the match between *localizing using an in-focus standard PSF model* and observed images of a *defocused emitter using the standard PSF*, we detect defocus mismatches (beyond 100 nm) relatively well using WIF. We can detect these subtle defocus errors *despite the fact that the standard PSF has poor sensitivity for 3D localization*. We only showed defocus detection up to 300 nm as most 2D SMLM experiments

are confined within that axial range, but we expect larger defocus errors to be *successively easier* to detect. We stress that this observation does not preclude the use of WIF for a 3D PSF, e.g., the long range DH-PSF.

Figure 2: Quantifying position inaccuracy along the y axis via WIF for the astigmatic PSF and DH-PSF. Example of one of 200 simulated images of a molecule using the astigmatic PSF at (a) focus and (b) $z = 400$ nm. We manually added a localization error of 50 nm along y (equivalent to half of the camera pixel size of 100 nm) to the ground-truth position of molecule from (a) and (b) and fed this new position into the WIF algorithm. Resulting computed WIFs across 200 independent images at (c) focus and (d) $z = 400$ nm. (e-h) Similar to (a-d) but for the DH-PSF. For both 3D PSFs, we used a brightness of 3000 photons and a uniform background of 20 photons per pixel. In computing WIFs, we used a grid distance of 50 nm and a regularizer value of 0.1. Colorbars: photons/ 100×100 nm².

As described in the main text, the extension of WIF to 3D PSFs uses an “in-plane” perturbation strategy, which is equivalent to the strategy for 2D PSFs. In Fig. 2, we can see that if position estimates are inaccurate along the y axis by half of the camera pixel size (pixel size = 100 nm), then the computed WIFs for both the DH-PSF and astigmatic PSF are much smaller than 1. This inaccuracy is detected both when a molecule is located at focus ($z = 0$) and far away from focus ($z = 400$ nm). Therefore, WIF is a general technique for quantifying inaccuracies in the position estimates for various 3D PSFs.

Then, the results and analyses described in the Supplementary Note 6 and Fig. S6 may be different depending on the different approaches? If so, what is the best 3D imaging approach suitable for WIF?

This is a very interesting question. Indeed, WIF sensitivity and performance can be different for distinct PSFs. For example, the astigmatic PSF is more sensitive to localization errors along the y axis at $z = 400$ nm than is the DH PSF in terms of WIF (see the lower average WIF score for the astigmatic PSF in Figs. 2(d,h)). This higher sensitivity could stem from the larger symmetric footprint of astigmatic PSF, as opposed to the double-spot DH PSF,

at $z = 400$ nm; given our symmetric perturbation, a higher fraction of perturbed sources are observed to decrease the resulting WIF. Such a result is not completely intuitive, and we believe that the WIF perturbation scheme, SNR, and 3D PSF all affect WIF values. Determining an optimal 3D PSF for WIF warrants a separate investigation.

Reviewer 3

The authors answered in a very complete manner to my requests and questions. The manuscript has been improved from the reviewers' comments.

The authors provide a metric for each molecule without the need of ground-truth positions (as opposed to Jaccard Index, etc.). This manuscript offers a tool which was missing for real SMLM data.

One comment:

Their metric does not attempt to explain the source of low confidence, but the authors proposed diverse flavours of WIF (geometry of transport or others) to provide some insights.

Mainly, this behaviour is correct if the likelihood used by WIF is accurate. Otherwise, as shown by the authors, WIF can yield an opposite scenario if the noise model is inaccurate. Another aspect is that some training data are probably needed to calibrate the hyper-parameters for new applications, although the authors suggest that their choice of hyper-parameters is fairly robust.

These aspects suggest that some expertise and carefulness are required for its usage and interpretation, but this is also the case for FRC/FSC when applying on real SMLM data.

Globally, the revised paper is clearer and more honest regarding the cautions to take, and the authors assess a large panel of scenarios. The additional experiments also show several behaviours of WIF (e.g., wrt regularizer strength) and how to take advantage of them. These will help the readers to better understand the proposed tool.

In the spirit of open science, I strongly encourage the authors to publish a usable / user-friendly software. The documentation should also state the limitations and/or cautions the user should take. That would only help for its dissemination and usage.

In conclusion, I support the publication of the submitted paper.

We thank the reviewer for their honest and thorough assessment of our work. We agree and have prepared detailed documentation of our MATLAB code. In addition, a thorough example walk-through of a WIF analysis workflow is provided to clarify how the software is to be operated. These materials can be found at https://osf.io/d72zv/?view_only=f7b47b8b542246d1a326d8b8a8c3a60f, as stated in our data and code availability statement in the main text.

Moreover, we are collaborating with researchers from other universities to implement a Python version of our algorithm. We will soon make this version public on our GitHub repository (<https://github.com/Lew-Lab>) after software validation and testing.

References

- [1] Long F, Zeng S, Huang ZL (2012) Localization-based super-resolution microscopy with an sCMOS camera part ii: Experimental methodology for comparing sCMOS with emccd cameras. *Optics Express* 20(16):17741–17759.
- [2] von Diezmann A, Lee MY, Lew MD, Moerner W (2015) Correcting field-dependent aberrations with nanoscale accuracy in three-dimensional single-molecule localization microscopy. *Optica* 2(11):985–993.

- [3] Ding T, Wu T, Mazidi H, Zhang O, Lew MD (2020) Single-molecule orientation localization microscopy for resolving structural heterogeneities between amyloid fibrils. *Optica* 7(6):602–607.
- [4] Thompson RE, Larson DR, Webb WW (2002) Precise nanometer localization analysis for individual fluorescent probes. *Biophysical journal* 82(5):2775–2783.

Reviewers' Comments:

Reviewer #1:

Remarks to the Author:

All my questions have been answered. I support the publication.